# Incorporating gauge-invariance in equivariant networks

## Abstract

Gauge theories, which describe fundamental forces in nature, arise from the principle of locality in physical interactions. These theories are characterized by their invariance under local symmetry transformations and the presence of a gauge field that mediates interactions. While recent works have introduced gauge equivariant neural networks, these models often focus on specific cases like tangent bundles or quotient spaces, limiting their applicability to the diverse gauge theories in physics. We propose a novel architecture for learning general gauge invariant quantities by explicitly modeling the gauge field in the context of graph neural networks. Our framework fills a critical gap in the existing literature by providing a general recipe for gauge invariance without restrictions on the fiber spaces. This approach allows for the modeling of more complex gauge theories, such as those with $SU(N)$ gauge groups, which are prevalent in particle physics. We evaluate our method on classical physical systems, including the XY model on various curved geometries, demonstrating its ability to capture gauge invariant properties in settings where existing equivariant architectures fall short. Our work takes a significant step towards bridging the gap between gauge theories in physics and equivariant neural network architectures, opening new avenues for applying machine learning to fundamental physical problems.

## 1 Introduction

Gauge theories form the cornerstone of modern physics, providing a unified framework for describing fundamental forces in nature (Weinberg, 1995). The Standard Model of particle physics, our most successful theory of fundamental interactions, is built upon gauge theories: $U(1)$ for electromagnetism, $SU(2) \times U(1)$ for electroweak theory, or $SU(3)$ for strong interactions (quantum chromodynamics) (Peskin & Schroeder, 2018). These theories arise from a profound principle: the locality of physical interactions, leading to a geometric structure described by fiber bundles (Nakahara, 2018).

Gauge theories offer a geometric interpretation of forces. In this framework, forces are understood as the curvature of fiber bundles, with particles represented as "sections" of these bundles (Baez & Muniain, 1994) called a "field" in physics (roughly speaking, distributions). For instance, the electromagnetic field can be viewed as a "connection" (aka gauge field) on a principal $U(1)$ bundle, while the gravitational field in General Relativity can be interpreted as a connection on the frame (local coordinates) bundle of spacetime (Frankel, 2011).

Despite their success, current physical models face limitations. The Standard Model, while extraordinarily accurate in its predictions, fails to incorporate gravity and cannot explain phenomena like dark matter or dark energy (Olive et al., 2014). These challenges motivate the exploration of more general models that maintain crucial symmetries while offering greater flexibility. Gauge invariance stands as a fundamental principle in physics, not merely a mathematical convenience. It embodies the idea that physical laws should be independent of our choice of local reference frame, much like how the laws of physics should be the same in all inertial reference frames (Zee, 2010). This principle leads to conservation laws through Noether's theorem, connecting symmetries to conserved quantities (Kosmann-Schwarzbach et al., 2011).

The development of gauge-equivariant neural networks opens exciting possibilities for discovery in fundamental physics. By constructing models that respect gauge symmetries, we not only improve

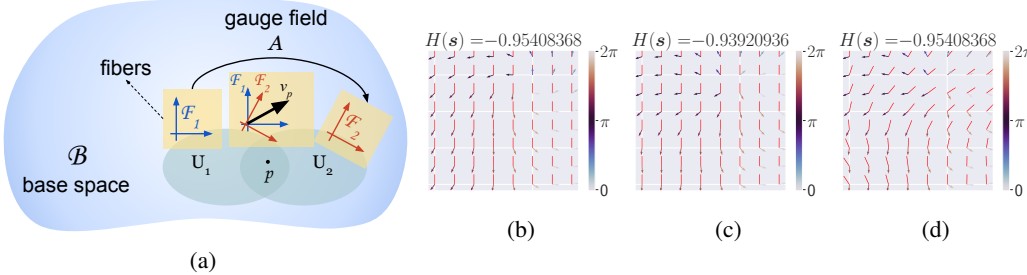

(b)    (c)    (d)

(a)

Figure 1: **Fiber bundles and gauge invariance.** (a) Overlapping neighboring open sets $U_1, U_2 \subset \mathcal{B}$ can have different bases in the fibers $\mathcal{F}_1$, $\mathcal{F}_2$ to represent *the same* object $v_p$ at point $p \in U_1 \cap U_2$. A gauge field $A$ accounts for such local (gauge) basis transformations in the fibers, ensuring measured quantities are invariant under them. Examples of gauge invariance: (b-d) The energy of locally interacting magnets (XY model) changes little by small perturbation to spins and is completely invariant under local basis changes.

computational efficiency but also align our tools with the underlying principles of nature. This approach could lead to new insights into multi-scale phenomena, help in exploring generalizations of current theories, and potentially uncover new physics beyond the Standard Model (Carleo et al., 2019).

In recent years, the machine learning community has made significant strides in incorporating symmetries and invariances into neural network architectures. Group equivariant convolutional neural networks (G-CNNs) (Cohen & Welling, 2016; Dieleman et al., 2016; Cohen & Welling, 2017) and gauge equivariant mesh CNNs (de Haan et al., 2021) have demonstrated the power of building known symmetries into model architectures. These approaches have shown remarkable improvements in tasks ranging from image classification to 3D shape analysis.

However, existing equivariant neural network architectures often fall short of capturing the full richness of gauge theories as they appear in physics. Many current approaches are limited to specific geometric settings, such as homogeneous spaces or quotient spaces (Kondor & Trivedi, 2018; Cohen et al., 2019a), or focus on cases where the same group acts on both the base space and the fibers (Weiler et al., 2018; Weiler & Cesa, 2020). While these simplifications have yielded positive results in certain domains, they are inadequate for modeling the complex gauge theories that describe fundamental physical interactions, where the fiber spaces can have diverse structures.

A critical component missing from most current equivariant neural network formulations is an explicit representation of the gauge field. In physical gauge theories, the gauge field mediates interactions and encodes the geometric structure of the fiber bundle. Its absence in current machine learning models limits their ability to capture the full complexity of gauge-theoretic systems.

In this paper, we address these limitations by proposing a novel architecture that explicitly models the gauge field within the framework of graph neural networks. Our approach focuses on local invariance under fiber symmetry groups, without assuming a direct correspondence between base space and fiber symmetries. By introducing a gauge correction term to align fibers at different points, we take a significant step towards modeling general gauge theories in a machine learning context. Specifically, our contributions are:

1. A generalized formulation of gauge equivariant neural networks that can handle arbitrary fiber bundles, moving beyond the limitations of current approaches restricted to tangent bundles or principal bundles (Cohen et al., 2019b; de Haan et al., 2021).
2. Explicit incorporation of the gauge field into the neural network architecture, allowing for the modeling of complex gauge theories as in physics (Luo et al., 2022; 2023).
3. Application of our approach to classical physical systems, including the XY model on various curved geometries, showcasing its ability to capture gauge invariant properties where existing equivariant architectures fall short (Chen et al., 2022; Bogatskiy et al., 2020).

We aim to open new avenues for applying machine learning to fundamental physical problems and potentially leading to more powerful and physically informed machine learning models.

## 2 BACKGROUND AND MOTIVATION

This section aims to provide intuition for gauge invariance and introduce the concept of gauge fields through concrete examples, progressing from simple to more complex ideas.

In both physics and applied mathematics, the concept of a "field" is fundamental to understanding how properties vary across space. In simple terms, a field is a state variable that depends on spatial coordinates. For instance, in an image, each pixel $p \in \mathbb{Z}^2$ has a color $v_p \in \mathcal{F} = [0,1]^3$ with values in the feature space $\mathcal{F}$. This construction where to each neighborhood of a base space (e.g. $\mathbb{Z}^2$) we attach a feature space (called "fibers") is an example of a trivial "fiber bundle" (Baez & Muniain, 1994) which we will formally define later. This notion can be extended to more complex data structures, like temperature or wind direction across a region, electromagnetic fields in physics, or any other variable distributed over a spatial domain.

**Intuition for gauge invariance.** In the example of images, the basis for the fibers needs to be the same for all of the fibers to ensure colors are defined consistently. However, in many problems in physical sciences, the quantities of interest may be invariant under local basis transformations, called "gauge transformations". Gauge theory is a paradigm for modelling such systems. A simple analogy can be drawn from the field of image processing. For instance, consider the task of digit recognition in MNIST. When these images undergo small local distortions–small diffeomorphisms–the identity of the digits remains unchanged.

This idea of invariance under local transformations extends beyond image processing to fundamental physical theories. To illustrate this concept more concretely, we now turn to a simple physical system that exhibits gauge invariance.

**Example: XY model.** In physics, a clear illustration of gauge theory principles can be seen in the XY model (Nagaosa, 2013), a system of 2D magnets (spins) interacting on a lattice (Figure 1c). [1]. Let $\boldsymbol{s}_j = \exp[i\theta_j] \in \mathbb{C}$ (with $\theta_j \in [0, 2\pi]$) denote the continuous spin state at node $j \in \mathcal{B} = \{1, \ldots, N\}$ of the lattice. Here, the lattice $\mathcal{B}$ is the "base space" and $\mathbb{C}$ at each $j$ where the $\boldsymbol{s}_j$ take values are the "fibers". Given the overall spin configuration $\boldsymbol{s} = \{\boldsymbol{s}_i\}_{i=1}^N$, the energy is defined as

$$H(\boldsymbol{s}) = -\sum_{ij} J_{ij} \boldsymbol{s}_i^T \boldsymbol{s}_j = \frac{1}{2} \sum_{ij} J_{ij} \|\boldsymbol{s}_i - \boldsymbol{s}_j\|^2 - N \tag{1}$$

where $J$ is the coupling matrix (the adjacency matrix of the lattice in this case). Clearly, the XY system's energy is invariant under *global* rotation in the fiber spaces $\boldsymbol{s}_j \rightarrow e^{i\alpha} \boldsymbol{s}_j$ (the same transformation on all the fibers). But, importantly, it also is invariant under *local* rotations, where in different neighborhoods of the lattice, the neighboring fibers are rotated together in such a way that $\boldsymbol{s}_i^T \boldsymbol{s}_j$ is not changed. This local symmetry is the "gauge invariance" of the XY model.

To allow for the fiber basis at each point to transform independently, we need to introduce a "gauge field" $A$. The gauge field is crucial because it cancels the difference in gauge (basis) at neighboring points, ensuring that the physical observables remain invariant under local transformations. Without the gauge field, the system would not be able to maintain its invariance under local rotations of the spins. To incorporate the gauge field, we redefine the energy as

$$H(\boldsymbol{s}, A) = \frac{1}{2} \sum_{ij} J_{ij} \|(1 + A_{ij})\boldsymbol{s}_i - \boldsymbol{s}_j\|^2 - N \tag{2}$$

and define the local transformations of $\boldsymbol{s}$ and $A$ as (see Appendix B)

$$\boldsymbol{s}_j \rightarrow e^{i\alpha_j} \boldsymbol{s}_j,$$
$$A_{ij} \rightarrow e^{i(\alpha_j - \alpha_i)}(1 + A_{ij}) - 1. \tag{3}$$

This formulation ensures that the energy of the system remains invariant under local rotations of the angle defining the spins. The gauge field $A$ adjusts to compensate for the local transformations, maintaining the overall invariance of the system. This observation is essential for designing neural network architectures that can recognize and respect local symmetries and transformations. By incorporating gauge fields into our models, we can create more flexible and powerful architectures capable of capturing the rich structure of gauge-invariant systems.

---

[1]In Appendix C.1 we provide further context as to how Figure 1b is generated.

**Continuous XY model.** To connect our discrete model to continuum physics, we now consider the continuous limit of the XY model. In this case, instead of a discrete lattice, the spins are defined as a continuous complex field $S : \mathbb{R}^2 \to \mathbb{C}$, where the base space is $\mathcal{B} \sim \mathbb{R}^2$ and the fibers are the same as before. In this continuous formulation:

- The discrete position index $i$ becomes a continuous variable $x$.
- The neighboring sites $j$ are now represented by $x + \delta x$, where $\delta x$ is infinitesimal.
- Expanding to first order in $\delta x$, we find $\boldsymbol{s}_i - \boldsymbol{s}_j = -\delta x \cdot \nabla S(x)$.
- The discrete gauge field $A_{ij}$ becomes a continuous function $A(x)$, where $A_{ij} \to \delta x A(x)$.

With these transformations, the energy of the system can be expressed as a continuous integral:

$$H = \int dx^2 \|(\nabla + A)S(x)\|^2 \tag{4}$$

The gauge transformation in this continuous setting takes the form:

$$S(x) \to e^{i\alpha(x)} S(x), \qquad\qquad A \to A - i\nabla\alpha \tag{5}$$

These transformations keep the overall energy unchanged. This illustrates a fundamental concept in gauge theory: gauge fields absorb local changes in the fiber space, ensuring that global properties, like energy, remain consistent under local transformations. This continuous formulation provides a bridge to more complex gauge theories in physics, where fields and their interactions are typically described by continuous functions over spacetime.

## 3 THEORY

A key principle in developing equivariant neural networks is the construction of features that transform predictably under the symmetry groups of the system. This approach has been successfully applied in various contexts, from Euclidean symmetries to more complex group structures (Cohen & Welling, 2016; Weiler & Cesa, 2020; Satorras et al., 2021). The general idea is to use objects that transform under different representations of the relevant symmetry groups to craft equivariant features. These objects, which we will formally define as tensors below, can be of various orders, each transforming linearly under a specific representation of the group. For instance, scalars (order-0 tensors) are invariant under group transformations, while vectors (order-1 tensors) transform under the fundamental representation of the group. Higher-order tensors can capture more complex transformation properties.

### 3.1 GAUGE SYMMETRY

**Differential geometry** Informally, a manifold is a space that locally resembles $\mathbb{R}^n$. We define an $n$-*dimensional manifold* $\mathcal{B}$ as a topological space equipped with charts (local mappings) $f_\alpha : U_\alpha \to \mathbb{R}^n$, where $U_\alpha \subset \mathcal{B}$ are open sets that cover $\mathcal{B}$. These charts are such that the transition function $f_\alpha \circ f_\beta^{-1}$ is smooth where it is defined.

**Definition 3.1** (Fiber bundle). A fiber bundle is a triple $\mathcal{E} = (\mathcal{B}, \mathcal{F}, \pi)$ consisting of a total space $\mathcal{E}$, a base space $\mathcal{B}$, and a projection map $\pi : \mathcal{E} \to \mathcal{B}$. For each point $p \in \mathcal{B}$, the set

$$\mathcal{F}_p = \{q \in \mathcal{E} : \pi(q) = p\} \tag{6}$$

is called the fiber over $p$. A "*section*" of the fiber bundle is a map $\phi : \mathcal{B} \to \mathcal{E}$ such that for any point $p \in \mathcal{B}$, it holds that $\pi(\phi(p)) = p$.

In physics, sections are called *fields*. A **trivial fiber bundle** is a particular case where the total space $\mathcal{E}$ is simply the product of the base space $\mathcal{B}$ and the fiber $\mathcal{F}$. For instance, an image is a section of a trivial fiber bundle. In the problems we are interested in, the fiber space has a symmetry group which acts on it. A **principal bundle** is a special type of fiber bundle particularly significant in gauge theories. In a principal bundle, each fiber is a homogeneous space—an orbit of a group $G$ acting smoothly, freely, and transitively.

Gauge theories primarily deal with principal bundles, as they are crucial for defining connections (gauge fields). These connections describe how to "transport" elements within the fiber from one point in the base space to another.

**Connections in Fiber Bundles**   A connection [2], or gauge field, on a fiber bundle is a mathematical tool that allows us to define how to "transport" elements along paths within the base space $\mathcal{B}$, while respecting the fiber structure. As illustrated in Figure 1, different local bases in the fibers leads to conflicting representations of fields. A connection enables us to compare elements of the fiber over different points in $\mathcal{B}$, effectively defining a "covariant derivative". The key idea is that the standard derivative $\partial_\mu$ does not account for the difference in the basis of fiber $\mathcal{F}_p$ and $\mathcal{F}_{p+\delta x}$. A gauge field $A$ modifies the standard derivative $\partial_\mu$ to a covariant derivative $D_\mu$ to maintain consistency between the local frames.

$$D_\mu = \partial_\mu + A_\mu. \tag{7}$$

**Gauge equivariance**   Let $G$ denote the the symmetry group of all $\mathcal{F}_p$, and let $g : \mathcal{B} \to G$ be set of group transformations which can act on a section (field) $\phi : \mathcal{B} \to \mathcal{E}$ to yield $\phi'(p) = g(p)\phi(p)$. To achieve gauge equivariance, the covariant derivative needs to be gauge (or group) equivariant, i.e., $D'(g \cdot f) = g \cdot Df$. This implies a constraint on the vector potentials of the form

$$A' = gAg^{-1} - (\nabla g)g^{-1}. \tag{8}$$

This concept is fundamental in physics and differential geometry, as it enables us to compare elements of the fiber over different points in $\mathcal{B}$, defining a "covariant derivative".

In a more formal setting, $A_\mu$ can be seen as a one-form on the base manifold that takes values in the Lie algebra of the gauge group $G$. This Lie algebra-valued one-form is defining how the gauge transformations adjust the fields and their derivatives, ensuring all physical predictions remain invariant under local transformations of the gauge group.

**Gauge arising from embedding**   Consider a fiber bundle $\mathcal{E} = (\mathcal{B}, \mathcal{F}, \pi)$ and a local trivialization[3] $\psi : \pi^{-1}(U) \to U \times \mathcal{F}$ over an open set $U \subset \mathcal{B}$. For a point $p \in U$, we can define the local map $\psi_p$ as the restriction of $\psi$ to $p$

$$\psi_p : \mathcal{F}_p = \pi^{-1}(p) \to \{p\} \times \mathcal{F} \cong \mathcal{F}. \tag{9}$$

Then, the pushforward $\psi_*$ between two points $p, q \in U$ is a map with

$$\psi_*^{p \to q} : \mathcal{F}_p \to \mathcal{F}_q \tag{10}$$

$$\psi_*^{p \to q} = \psi_q^{-1} \circ \psi_p. \tag{11}$$

This allows us to express objects between fibers at different points in a consistent way. In the case where the $\psi_p$ is a linear map, we can represent it as a matrix $\boldsymbol{E}_p$. Then, for $\alpha \in \mathbb{R}^d$ (assuming $d$-dimensional fibers):

$$\psi_p(\alpha) = \boldsymbol{E}_p \alpha. \tag{12}$$

In such a case, $\boldsymbol{E}_p$ represents the local frame at point $p$. The gauge field $A_{ij}$ can then be expressed in terms of the pushforward operator:

$$A_{ij} = \psi_*^{i \to j} - I = \psi_j^{-1} \circ \psi_i - I = \boldsymbol{E}_j^{-1} \boldsymbol{E}_i - I \tag{13}$$

In the special case where $\boldsymbol{E}$ is a rotation matrix (i.e., $\boldsymbol{E} \in SO(n)$), we have $\boldsymbol{E}^{-1} = \boldsymbol{E}^T$, and the expression simplifies to:

$$A_{ij} = \boldsymbol{E}_j^T \boldsymbol{E}_i - I \tag{14}$$

This formulation provides a more general framework for understanding how the gauge arises from the embedding, which reduces to the specific case described in the original text when the trivialization map is linear or when it is a rotation.

---

[2]The term "connection" is used slightly differently among mathematicians, who use it for the covariant derivative $D$, compared to physicists, using it for the gauge field $A$. To avoid confusion we will mostly use the terms covariant derivative for $D$ and gauge field for $A$.

[3]Meaning, mapping a neighborhood of the fiber bundle to a trivial bundle, i.e. a Cartesian product.

**Embedded XY**  As discussed in Section 3, embedding can induce a natural gauge on the fiber bundle. The XY model on a sphere provides a concrete example of this phenomenon. In this case, we identify the fibers with the tangent spaces, such that $\mathcal{F}_p \simeq T\mathcal{B}_p \simeq \mathbb{R}^2$.

When we map the XY system to a sphere in $\mathbb{R}^3$, we naturally induce a nontrivial gauge. To understand this, let's consider the local trivialization $\psi : U \subset S^2 \to \mathbb{R}^2$ using spherical coordinates $(\varphi, \theta)$, where $\varphi \in [0, 2\pi)$ is the azimuthal angle and $\theta \in (0, \pi)$ is the polar angle.

The basis vectors $\partial_{(\varphi,\theta)(p)} \equiv (\partial/\partial\varphi(p), \partial/\partial\theta(p))$ form the local frame $\boldsymbol{E}_p$ for $T\mathcal{B}_p$, as described in our theory section. To see how this frame varies from point to point, we can express it in terms of the embedding coordinates $\vec{p} = (x, y, z)$:

$$\boldsymbol{E}_p = \frac{\partial(x, y, z)}{\partial(\varphi, \theta)} = J \tag{15}$$

where $J$ is the Jacobian of the transformation. This Jacobian plays the role of the linear map $E$ in our general theory, allowing us to compute the gauge field $A_{ij}$ between neighboring points $i$ and $j$ as:

$$A_{ij} = J_j^{-1} J_i - I. \tag{16}$$

Notably, despite the nonlinear relation between Cartesian and spherical coordinates globally, the Jacobian provides a linear map between tangent spaces locally, allowing us to apply our linear gauge field formalism.

This formulation directly connects to our theoretical framework, demonstrating how the embedding induces a gauge field on the sphere. The non-trivial nature of this gauge arises from the curvature of the sphere, which prevents a global, consistent choice of basis for the tangent spaces.

**Curvature from Parallel Transport**  To understand the concept of curvature in gauge theories, let's consider how a field (section) $f(x)$ changes as we transport it around a small closed loop in the base space. We can describe this change using the covariant derivative $D_\mu = \partial_\mu + A_\mu$, where $A_\mu$ is the gauge field.

Consider a small rectangular loop in the x-y plane with sides $\delta x$ and $\delta y$. As we move around this loop, the field transforms as:

$$f \xrightarrow{D_x \delta x} (1 + D_x\delta x)f \xrightarrow{D_y\delta y} (1 + D_y\delta y)(1 + D_x\delta x)f$$
$$\xrightarrow{-D_x\delta x} (1 - D_x\delta x)(1 + D_y\delta y)(1 + D_x\delta x)f$$
$$\xrightarrow{-D_y\delta y} (1 - D_y\delta y)(1 - D_x\delta x)(1 + D_y\delta y)(1 + D_x\delta x)f \tag{17}$$

Expanding to second order and simplifying, we find:

$$f \to f + ([D_x, D_y]\delta x\delta y)f + O(\delta^3) \tag{18}$$

The commutator $[D_x, D_y]$ defines the curvature $F_{xy}$:

$$F_{xy} = [D_x, D_y] = (\partial_x A_y - \partial_y A_x + [A_x, A_y]) \tag{19}$$

This is the xy-component of the curvature tensor $F_{\mu\nu}$. In the case of an Abelian gauge theory like $U(1)$, the commutator $[A_x, A_y]$ vanishes, simplifying the expression further.

### 3.2 Gauge Equivariant Graph Neural Network Architecture

To implement our gauge-equivariant neural network, we need to discretize the fiber bundle and adapt the message passing framework of Graph Neural Networks (GNNs) to incorporate gauge equivariance. Here, we present a general formulation of our architecture.

**Discretization of the Fiber Bundle**  We encode a simplified topology of the base space $\mathcal{B}$ as a graph $\mathcal{G} = (\mathcal{V}, \mathcal{C})$, where $\mathcal{V}$ represents the set of vertices and $\mathcal{C} \subset \mathcal{V} \times \mathcal{V}$ the set of edges. This graph structure approximates the topology of $\mathcal{B}$, forgoing the full simplicial complex representation for computational efficiency.

In this discretized setting, the gauge field $A$, which encodes the transformation between local frames, is represented as a function on the edges of the graph:

$$A : \mathcal{C} \to \text{Hom}(\mathcal{F}), \tag{20}$$

where $\text{Hom}(\mathcal{F})$ denotes the space of linear transformations on the fiber space $\mathcal{F}$. For each edge $(i, j) \in \mathcal{C}$, $A_{ij}$ represents how the coordinates of frames $\mathcal{F}_i$ and $\mathcal{F}_j$ map to each other.

**Node and Edge Features**   In our gauge-equivariant GNN, the node features differ from traditional GNNs to respect gauge symmetry:

1. **Vector Features**: The value of the field (section) at each node is represented as $s_i \in \mathcal{F}_i$.

2. **Scalar Features**: We can compute gauge-invariant scalar features at each node, such as $\|s_i\|$, using a metric and inner product defined on the fiber space $\mathcal{F}_i$.

3. **Mixed Features**: Unlike traditional GNNs, we cannot directly use $s_i s_j$ as a feature, as it is not gauge equivariant or invariant. Instead, we use the gauge field $A_{ij}$ to construct gauge-equivariant features:

a. Scalar mixed features: $\langle (I + A_{ij})s_i, s_j \rangle$

b. "Bivector" features: $(I + A_{ij})s_i - s_j$, which is the discrete analog of the covariant derivative $Ds(x)$.A

**Message Passing**   In our gauge-equivariant message passing scheme, messages from node $i$ to node $j$ are transformed using the gauge field $A_{ij}$. The message passing operation can be generalized as

$$m_{i \to j} = \phi\left(s_i, s_j, A_{ij}, \langle (I + A_{ij})s_i, s_j \rangle, (I + A_{ij})s_i - s_j\right), \tag{21}$$

where $\phi$ is a learnable function that respects gauge equivariance.

**Node Update**   The node update function aggregates incoming messages and updates the node's features in a gauge-equivariant manner:

$$s_i^{(t+1)} = \psi\left(s_i^{(t)}, \sum_{j \in \mathcal{N}(i)} m_{j \to i}\right), \tag{22}$$

where $\psi$ is another learnable function that preserves gauge equivariance, and $\mathcal{N}(i)$ denotes the neighborhood of node $i$.

**Geometric Interpretation**   The "bivector" feature $(I + A_{ij})s_i - s_j$ can indeed be interpreted as an element of $T\mathcal{B}_x \otimes \mathcal{F}_x$. More precisely, it is a section of the bundle $T\mathcal{B} \otimes \pi^* \mathcal{F}$, where $\pi^* \mathcal{F}$ is the pullback of the fiber bundle $\mathcal{F}$ to the total space $\mathcal{E}$. This interpretation aligns with the geometric nature of the covariant derivative in differential geometry.

While our current implementation focuses on the gauge field $A_{ij}$, future extensions could incorporate the curvature $F = [D, D]$ as additional features, potentially capturing higher-order geometric information of the fiber bundle.

## 4 EXPERIMENTS

As we discussed in Section 2, the XY model can be expressed as a gauge field and its energy is invariant to local (and global) transformations. We propose an architecture that constructs features that are, by design, gauge-invariant. Using the gauge-invariant features, we make local energy predictions, which are then aggregated to solve the problem of energy estimation in the XY model.

**Datasets**   We consider three datasets to showcase the necessity of our method in manifolds with different topologies. Specifically, we consider:

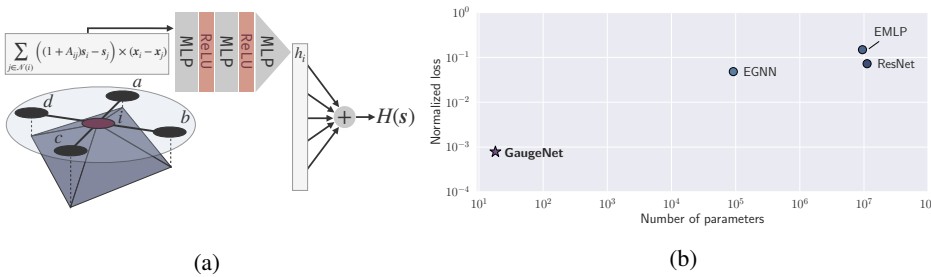

(a)             (b)

Figure 2: (a) Gauge-invariant architecture. Each neighborhood computes a gauge-corrected local estimate for the energy, which are then aggregated for the total energy of the configuration $H(\boldsymbol{s})$. (b) Performance of neural network architectures on energy regression for the XY model. Our method, GaugeNet, is shown in bold.

- fields over a grid on flat spaces, with boundary conditions,
- fields over toruses (which is isomorphic to the grid), and
- fields over spheres.

We present samples of low-, mid-, and high- energy configurations for all topologies in Figure 3. While the first and second datasets are isomorphic (and thus we expect them to produce similar configurations), we note that the torus has a curvature that makes direct comparison between the fibers nontrivial, and is thus a great example to showcase our method.

For each dataset we generated $N = 10,000$ samples, each representing a field configuration $\boldsymbol{s} \in \mathbb{R}^n$, where $n = 10,000$ denotes the number of discrete points. Original spin configurations were i.i.d. samples from a uniform distribution and iterated over $10,000$ gradient steps to reach each final configuration. The dataset consists of data pairs $\{(\boldsymbol{s}_i, y_i)\}_{i=1}^{10,000}$ with $y_i = H(\boldsymbol{s}_i) \in \mathbb{R}$ being the energy of each final configuration, which we consider as the target variable for a regression task. The XY model has multiple energy basins that the system can converge to. For details about the data-generating procedure see Appendix C.2 and for more samples from the grid dataset see Appendix C.4.

**Tasks** We considered the performance of our architecture against baselines on two tasks. The first task is a regression problem, where the models estimate the energy of a configuration given the final state. The second task is a classification problem where the models estimate the number of vortices in the configuration given the final state. The vortices were estimated from the energy distributions of the datasets, for more details see Appendix C.3.

**Architecture** In Section 3 we discussed many possible feature types that are compatible with our framework. For the problem of energy estimation, we opted for bivector features of the form $(I + A_{ij})s_i - s_j$, as the energy is dependent on this quantity. To allow the model to utilize the location information when the XY model is evaluated on nontrivial topologies, we incorporate the point positions, similar to EGNN. This yields node features of the form

$$\sum_{j \in \mathcal{N}(i)} \left( (I + A_{ij})s_i - s_j \right) \times (\boldsymbol{x}_i - \boldsymbol{x}_j), \tag{23}$$

where $\times$ denotes the outer product. To model the local contribution of each neighborhood of the XY model configuration to the overall energy, we adopt a simple two hidden layer MLP to estimate the local energy. Then the local energies are summed, yielding the overall estimate for the configuration $H(\boldsymbol{s})$. An illustration of the architecture, which we call **GaugeNet**, is given in Figure 2a.

For the classification task, we add an MLP with a single hidden layer on top of the network that estimates the energy. As discussed in Appendix C.3, for small numbers of vortices the energy can determine the number of vortex pairs, leading to this design choice.

**Baselines** For baseline evaluation we compare against high-performing general architectures, equivariant neural networks, and equivariant graph neural networks. We use ResNet18 (He et al.,

2016) as a general, high-performing baseline. Because ResNets work with image data, we expect they would particularly struggle on graph-structured data. For the equivariant baseline, we compare against EMLP (Finzi et al., 2021) equipped with SO(2)-invariance. While EMLP has invariance properties, they are global and not local, which could fail to capture the interactions in the XY model. Finally, we compare against EGNN (Satorras et al., 2021), an equivariant graph neural network. While this model

## 4.1 ENERGY ESTIMATION

We evaluated the performance of all the architectures in an energy estimation task on a grid topology. We expect the baselines to perform most favorably in such a task as the space has no curvature and the gauge is zero.

In Figure 2b, we show the performance of the architectures relative to their number of parameters. We see that GaugeNet provides significant performance gains over baselines, reducing the loss by almost a factor of 100. At the same time, our model is parameter efficient, requiring almost $10,000\times$ less parameters than the next smallest model. EMLP struggles to provide accurate predictions as it models global, instead of local invariances and suffers from large parameter counts due to its full connectivity. While great general purpose architectures, ResNets do not have any local invariance properties to allow them to perform favorably. Finally, EGNN is more parameter efficient than the other baselines due to its graph structure and performs all computations locally. However, despite its equivariance properties, it does not exploit the problem structure adequately to accurately predict the configuration's energy, and still requires orders of magnitude more parameters than GaugeNet.

## 4.2 VORTICITY CLASSIFICATION

To evaluate the effectiveness of our architecture, we considered a higher-level task on a sphere topology. The number of vortices can be inferred from the energy of the configuration (for a moderate number of vortex pairs), however the nontrivial topology makes the energy estimation more challenging. We present the results in Table 1[4].

While both the ResNet and EMLP achieved high training accuracies (53.17% for the ResNet, 99.7% for EMLP), their testing performance lagged behind. This indicates that they did not learn to estimate the energy correctly, due to the curved topology, and overfit to the training data. In contrast, GaugeNet is able to achieve high classification accuracy due to its gauge-correcting capabilities that lead to accurate energy estimations.

|          | GaugeNet | ResNet | EMLP  |
|----------|----------|--------|-------|
| Accuracy | **85.92** | 23.55  | 24.36 |

Table 1: Test accuracies our proposed architecture GaugeNet and two baselines, ResNet and EMLP. GaugeNet greatly outperforms baselines who overfit the training data.

## 5 CONCLUSION AND DISCUSSIONS

In this work we presented GaugeNet to solve the energy estimation problem in the XY model, utilizing the problem's structure to construct gauge-invariant features. As inner products are preserved under gauge transformations, at each neighborhood we compute the inner products between the central node and its neighbors, which we use as features. As the energy is location-agnostic, our model makes local energy estimations at every neighborhood, which are aggregated for the final prediction. GaugeNet provides significant performance gains over baselines which include both equivariant and general purpose architectures, for only a small fraction of the parameters.

---

[4]EGNN was not included in this table as it failed to converge to a reasonable solution.

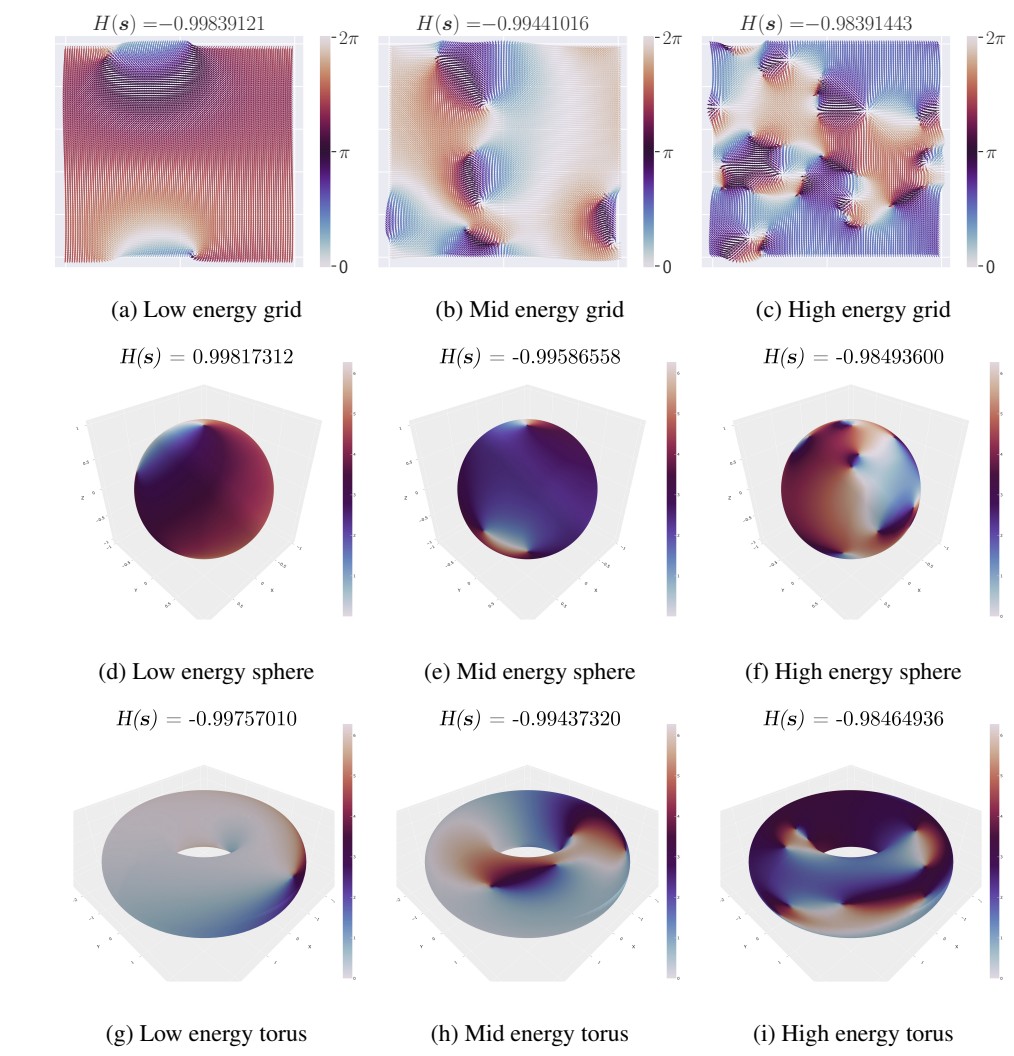

Figure 3: **XY models of varying energies in different topologies.** (a-c) Field topologies (isomorphic to toruses), (d-f) torus topologies, and (g-i) sphere topologies.

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

## A    RELATED WORK

Building upon the context provided in the introduction, we delve deeper into specific approaches and their limitations in equivariant neural networks and gauge theories.

**Equivariant feature construction.**    The development of equivariant neural networks has led to various approaches for constructing features that transform predictably under symmetry groups. These approaches can be broadly categorized based on how they handle different symmetries:
1. *Global symmetries:* Methods like Group Equivariant CNNs (Cohen & Welling, 2016) and Steerable CNNs (Cohen & Welling, 2017) focus on global symmetries, typically in the context of image processing. These approaches construct features that are equivariant to rotations, reflections, and other global transformations.
2. *Euclidean symmetries in graphs:* Equivariant Graph Neural Networks (EGNNs) (Satorras et al., 2021) extend equivariance to graph-structured data. They use combinations of scalar and vector features to achieve equivariance to Euclidean symmetries, which is particularly useful for molecular modeling and physical simulations.
3. *More complex symmetry groups:* Some works have tackled more complex symmetries, such as the Lorentz Group Network (LGN) (Gong et al., 2022), which constructs features equivariant to Lorentz transformations using the Clebsch-Gordan decomposition.

While these approaches have shown success in their respective domains, they typically assume a fixed, known symmetry group and often struggle with local symmetries or gauge transformations.

**Gauge equivariance approaches.**    Extending equivariance to gauge theories has presented unique challenges. Notable attempts include:

- Cohen et al. (2019b)'s work on gauge-equivariant CNNs for spherical surfaces, which handles the transition between local coordinate systems on a sphere.
- Gauge Equivariant Mesh CNNs (de Haan et al., 2021), which align representations across different points on discretized manifolds.
- Recent applications to quantum lattice models (Luo et al., 2023; Chen et al., 2022), demonstrating the potential of gauge-invariant networks in simulated physical systems.

However, these approaches often rely on specific geometric structures (like homogeneous spaces or tangent bundles) or assume simplified gauge groups. This limits their applicability to the diverse fiber bundle structures found in physical gauge theories.

Our work addresses these limitations by providing a more general framework for gauge equivariant neural networks. We extend the principle of equivariant feature construction to arbitrary fiber bundles by explicitly modeling the gauge field. This allows us to construct gauge-invariant features, which generalize the equivariant features used in previous approaches. By doing so, we provide a flexible framework that can handle more general gauge theories, particularly for discrete spaces and graph-structured data. This approach opens new possibilities for applying machine learning techniques to a wider range of physical systems governed by gauge theories, bridging the gap between theoretical physics and practical machine learning implementations.

## B   XY MODEL GAUGE SYMMETRY

In the XY model on a discrete lattice we defined the energy as

$$H(\boldsymbol{s}, A) = \frac{1}{2} \sum_{ij} J_{ij} \|(1 + A_{ij})\boldsymbol{s}_i - \boldsymbol{s}_j\|^2 - N \tag{24}$$

and define the local transformations of $\boldsymbol{s}$ as

$$\boldsymbol{s}'_j = e^{i\alpha_j} \boldsymbol{s}_j. \tag{25}$$

Here, we will derive how $A$ should transform such that the energy remains invariant. Thus, we want

$$H(\boldsymbol{s}, A) = H(\boldsymbol{s}', A') \tag{26}$$

We will match every term in the sum, such that for every pair $i, j$ we have

$$\begin{aligned}
\|(1 + A_{ij})\boldsymbol{s}_i - \boldsymbol{s}_j\|^2 &= \|(1 + A'_{ij})\boldsymbol{s}'_i - \boldsymbol{s}'_j\|^2 \\
&= \|(1 + A'_{ij})e^{i\alpha_i}\boldsymbol{s}_i - e^{i\alpha_j}\boldsymbol{s}_j\|^2 \\
&= \|e^{i\alpha_j}\left(e^{-i\alpha_j}(1 + A'_{ij})e^{i\alpha_i}\boldsymbol{s}_i - \boldsymbol{s}_j\right)\|^2 \\
&= \|e^{-i\alpha_j}(1 + A'_{ij})e^{i\alpha_i}\boldsymbol{s}_i - \boldsymbol{s}_j\|^2
\end{aligned} \tag{27}$$

From this we can equate the gauge parts and obtain

$$\begin{aligned}
1 + A_{ij} &= e^{-i\alpha_j}(1 + A'_{ij})e^{i\alpha_i} \\
1 + A'_{ij} &= e^{i\alpha_j}(1 + A_{ij})e^{-i\alpha_i} \\
A'_{ij} &= e^{i\alpha_j}(1 + A_{ij})e^{-i\alpha_i} - 1
\end{aligned} \tag{28}$$

which is shows how the gauge field transforms. We can readily verify that when all $\alpha_i = 0$, we recover $A' = A$. Also, note that since $A$ has to take values in the Lie algebra of the gauge group, $A_{ij} = M_{ij}L$, where $L = i = \sqrt{-1}$ is the basis of $\mathfrak{u}(1)$, the Lie algebra of $U(1)$. Since $U(1)$ is Abelian, $e^{i\alpha}A = Ae^{i\alpha}$ and so

$$A'_{ij} = e^{i(\alpha_j - \alpha_i)}(1 + A_{ij}) - 1. \tag{29}$$

For global transformations, where $\alpha_i = \alpha$, we again recover $A' = A$. This only happens for Abelian gauge symmetries.

## C   IMPLEMENTATION DETAILS

### C.1   EFFECT OF GAUGE TRANSFORMATIONS

In Section 2 we discussed that the spin configuration in the XY model changes as $\boldsymbol{s}_i \to e^{i\alpha_i}\boldsymbol{s}_i$ under a gauge transformation, where the gauge field $\alpha_i$ is smoothly changing. To generate the transformed XY configuration in Figure 1 we considered a simple gauge field that changes with a low frequency across the $x$ and $y$ dimensions. Specifically, let $\alpha_{i_x}, \alpha_{i_y}$ be the $x, y$ components of $\alpha_i$

$$\begin{aligned}
\alpha_{i_x} &= 0.25 \cdot \cos\left(2\pi \tfrac{x}{n}\right), \\
\alpha_{i_y} &= 0.25 \cdot \cos\left(2\pi \tfrac{2y}{n}\right),
\end{aligned}$$

where $n$ is the size of the grid. In this example, we assume the two components act independently on the spin configuration, and thus the transformed angle of each vector is $\theta_g = \theta + \alpha_{i_x} + \alpha_{i_y}$. The two components of $\alpha_i$ alongside the effective gauge field are shown in Figure 4.

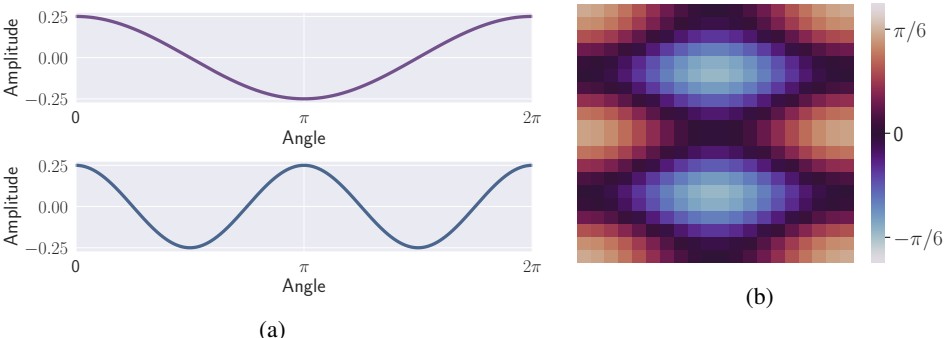

Figure 4: (a) Low-frequency components of the gauge field along the $x$ (top) and $y$ (bottom) directions. (b) Effective gauge field, comprising of an angular offset $\theta_g = \theta + \alpha_{i_x} + \alpha_{i_y}$.

## C.2 DATA GENERATION

Each data sample consists of a pair $(\boldsymbol{x}_i, y_i)$ with $\boldsymbol{x}_i \in \mathbb{R}^{100 \times 100}$ and $y_i \in \mathbb{R}$ denoting the spin configuration at each grid point and the energy of the configuration. Dropping the sample index for clarity, each $x_{kl} \equiv \theta_{kl}$ represents the angle of the spin vector at that grid point and $\boldsymbol{s}_{kl} = \begin{bmatrix} \cos(\theta_{kl}) \\ \sin(\theta_{kl}) \end{bmatrix}$.

Each angle is originally sampled as random variable $\theta_{kl} \sim 2\pi U$ with $U \sim \mathrm{Unif}$. We then run $\eta = 10,000$ gradient steps optimizing the energy of the configuration $H(\boldsymbol{s}) = -\sum_{ij} J_{ij} \boldsymbol{s}_i^T \boldsymbol{s}_j$.

The XY model has many low energy states and convergence to these states depends on the temperature of the system. In order to get a diverse distribution of energies suitable for a learning task, we need to sample different system temperatures. In order simulate different temperatures, we varied the learning rate of the optimizer $lr \in \{100, 10, 1, 0.1, 0.01\}$. Higher learning rates correspond to higher temperatures where the system is more likely to reach the ground state (where all charges are aligned). To ensure convergence of the descent procedure for all learning rates, we choose a target learning rate that showed consistent converging behavior ($lr_{\mathrm{final}} = 0.01$) and designed a linear learning rate schedule where the target is reached after $\kappa = 100$ learning rate updates. Specifically, we set $\gamma = \left(\frac{lr}{lr_{\mathrm{final}}}\right)^{\frac{1}{\kappa}}$ and we updated the learning rate every $\frac{\eta}{\kappa}$ gradient steps. We used the Adam optimizer and uniformly at random choose a learning rate for each data sample. The process to generate $N = 10,000$ samples took $\sim$10 hours on a NVIDIA GeForce RTX 3090 Ti.

## C.3 VORTEX ESTIMATION

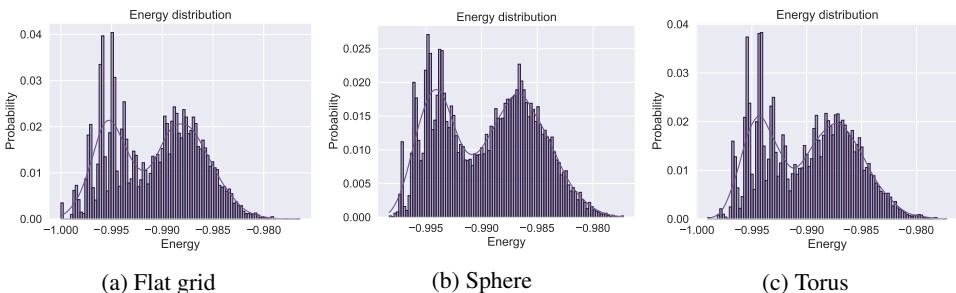

Figure 5: **Energy distributions of different datasets.** (a) Flat grid, (b) sphere, and (c) torus.

Due to the lack of ground truth information for the number of vortices and the difficulty of hand-labeling thousands of configurations, we opted for an automatic way to estimate them. Computing the energy distributions of each dataset (Figure 5), we observe a specific pattern: for all datasets, at low energies, the distributions are spiky, and after a critical energy, they become more normal, leading to an overall bimodal distribution.

The explanation for this phenomenon is that the energy contribution is local to each neighborhood and, when the vortices are small in number and far apart, each vortex contributes independently to the overall energy. The spikes at the low energies, then, correspond to configurations with distinct number of vortices. When the number of vortices increases, they inevitably also become closer and interactions are introduced between them. This makes estimating their number purely from the energy challenging when that number grows.

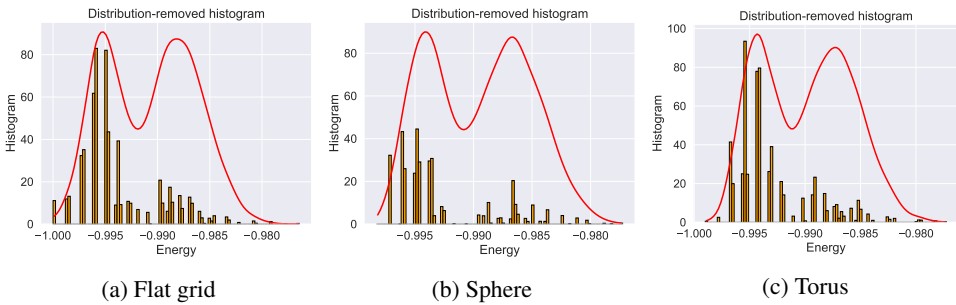

(a) Flat grid   (b) Sphere   (c) Torus

Figure 6: **Distribution-removed histograms.** (a) Flat grid, (b) sphere, and (c) torus.

Concretely, to estimate the number of vortices we first subtracted the estimated (smooth) distribution from the histogram in order to emphasize on the spikes. Then, we used `Scipy`'s `find_peaks` function to find *valleys* in the resulting histogram. We opted for valleys instead of peaks as they indicate the cutoffs between numbers of different vortices. This process is illustrated in Figures 6 and 7.

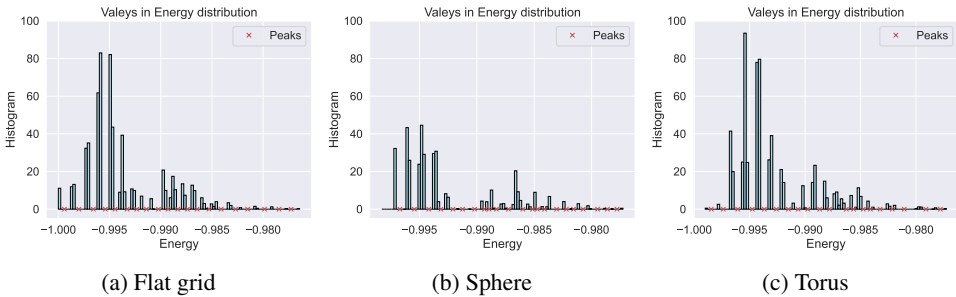

(a) Flat grid   (b) Sphere   (c) Torus

Figure 7: **Valleys in energy distributions.** (a) Flat grid, (b) sphere, and (c) torus.

By using the discrete energy cutoffs we can automatically label configurations with their number of vortices for classification purposes. However, we have to be aware of the phase transition where interactions between the vortices start to happen. To address that, we consider a critical number of vortices (in our experiments, 15) to be a catchall for any number of vortex pair larger than the critical number.

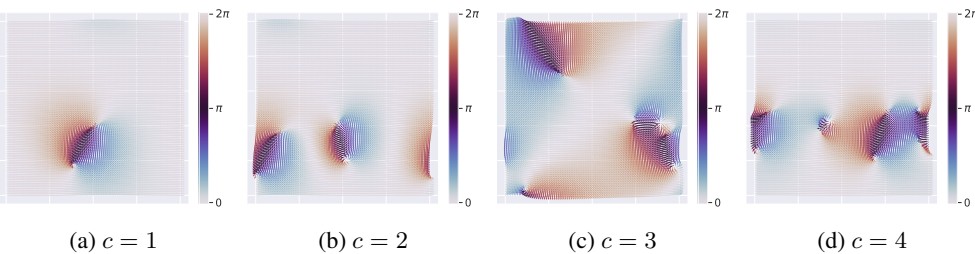

(a) $c = 1$  (b) $c = 2$  (c) $c = 3$  (d) $c = 4$

Figure 8: **Configurations of different vortex pairs.** (a) One pair, (b) two pairs, (c) three pairs, and (d) four pairs.

Finally, we include some samples of configurations with different numbers of vortices, as estimated by our algorithm, in Figure 8.

## C.4 XY MODEL SAMPLES

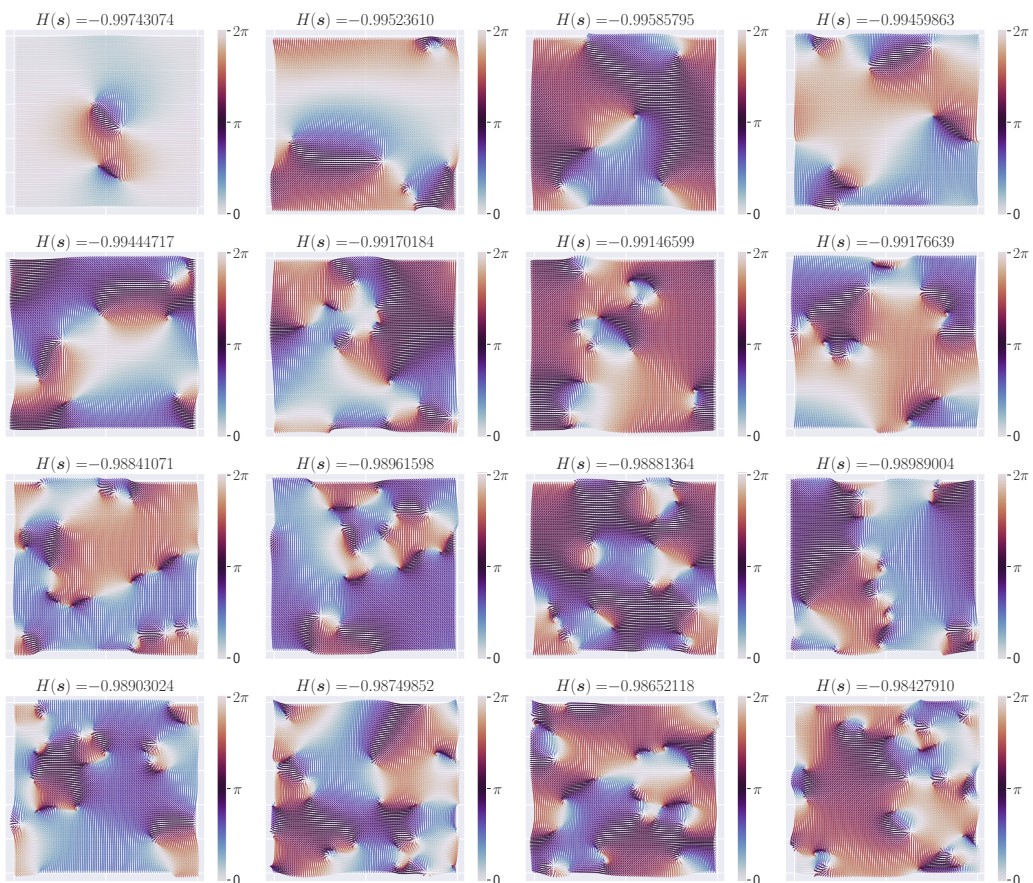

Figure 9: Final configurations in the XY model dataset alongside their energies (as the plot titles). Samples were picked at random from the full dataset ($N = 10,000$).

