# OpenReview forum: "Incorporating gauge-invariance in equivariant networks"
_ICLR.cc/2025/Conference — Submitted to ICLR 2025_

### Official Review · Reviewer_b1bp · 2024-11-02

**Soundness:** 1
**Presentation:** 1
**Contribution:** 2
**Rating:** 3
**Confidence:** 3

**Summary:**

After introducing foundational concepts such as fiber bundles, connections, gauge equivariance, and the XY model, the presented proposes GaugeNet, a graph neural network architecture specifically designed to manage gauge-invariant tasks by addressing local symmetries, which conventional equivariant models handle only at a global scale. Additionally, the authors propose a synthetic dataset designed for energy regression and vortex counting tasks on the XY model across different topologies. Finally, GaugeNet is evaluated on the proposed dataset using energy regression for toroidal topology and vortex counting for spherical topology.

**Strengths:**

* Highlights an important limitation of current equivariant methods that restrict their ability to model tasks with local symmetries.

**Weaknesses:**

- **Reproducibility Concerns:** The training procedures are insufficiently described, with no details provided on the dataset split for training, validation, and test sets, nor on the loss function, learning rate, or optimizer. Furthermore, I believe it is necessary to include an explanation of how the baselines were implemented for the given tasks or how they process the XY model configurations.
-  **Lack of Empirical Support for Comparison Claims:** The third contribution (L104) suggests that the authors apply their approach to classical physical systems to showcase the ability to capture gauge invariance, addressing properties that previous models do not account for (Chen et al., 2022; Bogatskiy et al., 2020). However, I believe it is necessary to validate this claim with empirical evidence by presenting an experimental comparison with these models.
- **Generalization Claim Not Supported:** The first contribution (L98) claims that the proposed model handles arbitrary fiber bundles and generalizes previous models restricted to principal bundles. However, all examples and the computational model presented are limited to vector bundles, which are a specific case of principal bundles, contradicting the claim. As an example, at L328 the gauge field $A$ is defined as a function mapping to linear endomorphisms of fibers. This supposes that the fiber is a vector space, hence the $\mathcal{E}$ is a vector bundle. To corroborate this claim and validate its effectiveness, I believe it is necessary to provide examples and experimental evidence on GaugeNet models defined on non-principal bundles.
- **Unclear and unpolished writing (No Impact on Recommendation):**

  I suggest that the authors carefully revise the text, as it contains numerous typos or inaccuracies. For example:

    - L394: "which is..."
    - L343: ".A"
    - L437: The text ends abruptly.
    - Fig 2a: "MLP" should be replaced by "linear layer" or similar.
    - L212: Group $G$ does not necessarily need to act freely on homogeneous spaces.

  In certain instances, the tone and language seem unsuited for academic purposes. For example:

    - L274, L295: "let us" should be used instead of the colloquial "let's".
    - L103: The sentence "Application of our approach to classical physical systems, including the XY model" is misleading since only the XY model is considered.

  The paper structure could be optimized by streamlining certain paragraphs and clarifying cross-references. For example:

    - L294-313: The discussion on curvature can be minimized, as it’s only briefly mentioned in L365-367 without further use.
    - L264: Unclear reference to "original text."

**Questions:**

1. What are the boundary conditions introduced at L393? Additionally, what is meant by "isomorphic" in the next line?
2. In L345-360, the gauge-equivariant message-passing scheme involves learnable functions $φ$ and $ψ$ (L349, L359). How are these functions constructed, and how do they respect gauge equivariance, in particular for bivector features?
3. Why was energy regression tested only on grid topology and vorticity classification only on spherical topology?
4. Why was ResNet18 chosen as a baseline, given that it is a model designed for image processing—a class of tasks quite different from the one it is trained on in this paper?
5. At L240, what does the term "embedding" in the paragraph title refer to? Is it related to local trivialization?

---

> ### Author Response · Authors · 2024-12-02
>
> We thank the reviewer for their thorough and constructive feedback, which has provided valuable guidance for improving our work. Below, we address the specific concerns raised.
>
> 1. **Reproducibility Concerns**:
>     - We acknowledge the need to include detailed descriptions of the training procedures. We omitted these details in the main text for clarity, while also following the standard practice of deferring technical details to code repositories (to be uncommented upon deanonymization). Sections C.2 and C.3 of the appendix already describe the data generation and vortex estimation processes, but as requested we will include an additional appendix section providing details on dataset splits (training/validation/test), loss functions, learning rates, optimizers, and other training parameters in an updated version.
>     - Regarding the baselines, we used the official implementations from their respective repositories. For ResNets, which are not designed to handle graph data, we reshaped point clouds into compatible image formats and for EMLP we flattened the data into vectors. This adaptation allowed us to test whether these models could solve the tasks, even without spatially structured information. We will clarify this process in an updated manuscript and will include details in the appendix.
>
> 2. **Comparison with Chen et al. and Bogatskiy et al.**: We appreciate the suggestion to include comparisons with Chen et al. (2022) and Bogatskiy et al. (2020). These methods are designed for global equivariance rather than local (gauge) transformations, making direct comparisons challenging. For example:
>     - Chen et al. study a quantum field theory, which isn't applicable for the XY model.
>     - Bogatskiy et al. focus on Lorentz-equivariant architectures, requiring datasets of relativistic particles. These models operate under global Lorentz transformations, which differ fundamentally from the local symmetries addressed by our approach.
>
>     Adapting these models to our setting would require significant modifications and is a substantial undertaking that could form the basis of a separate study. Our motivation in including these references was to highlight the difference in approach instead of providing a direct computational comparison. We believe our work complements these approaches by addressing gauge equivariance explicitly, which is not the focus of these methods. We will clarify these distinctions in the revised manuscript.
>
> 3. **Generalization Claim and Fiber Bundles**: Thank you for pointing this out. To clarify, while vector bundles can be associated with principal bundles, they are not strictly a special case of them. Principal bundles require the base manifold to be a quotient group, whereas vector bundles allow arbitrary manifolds as bases, with fibers being vector spaces. This distinction is supported by literature [1,2]. We will revise the text to ensure this distinction is more precise and include examples of how our method generalizes to broader applications.
>
> 4. **Typos, Writing Style, and Structure**: We appreciate the detailed feedback on writing clarity and structure. As also noted by Reviewer 4fJP, we will streamline sections that are less central to the results (e.g., curvature discussion) and refine our language for consistency and academic tone. Specific issues such as “MLP” in Figure 2a, typographical errors, and colloquial expressions will also be corrected in an updated version. Regarding the free action of groups on homogeneous spaces (L212), while it is not strictly required in general contexts [2], it is frequently assumed in the context of equivariant architectures [1]. However, we will clarify the phrasing for precision and avoid any potential ambiguity.

---

> ### Author Response · Authors · 2024-12-02
>
> 5. **Boundary Conditions and Isomorphism (L393)**: The grid configuration for the XY model uses periodic boundary conditions, where the top-most points neighbor the bottom-most, and similarly for left-right edges. This effectively makes the grid configuration isomorphic to a torus, though represented in a flat topology. This choice ensures vortical interactions while avoiding boundary artifacts. Alternatively, one could impose strong charges at the boundaries to create “charge walls,” but boundary conditions are standard for maintaining physical consistency.
>
> 6. **Learnable Functions and Gauge Equivariance (L345-360)**: In our experiments, the aggregate message function $\phi$ was given by Equation (23) and was not treated as a learnable function. The function $\psi$ was modeled by the MLP shown in Figure 2a. By construction, the gauge-correction mechanism ensures that all features remain gauge-invariant, maintaining the overall equivariance of the architecture.
> We abstracted these details in the general framework presented in this section to focus on the broader applicability of our method. We will clarify this abstraction and provide additional implementation details in the revised manuscript.
>
> 7. **Topology-Task Relationships**: We ran all experiments across all topologies but reported results for one topology per task for clarity and conciseness, as performance was consistent across different topologies. We will include results for all topologies in the appendix to demonstrate the generalizability of our method.
>
> 8. **Choice of ResNet18 as a Baseline**: ResNet18 was chosen as a baseline for two reasons:
>     - *General-purpose architecture*: ResNets are commonly used as starting points for investigations and have been adapted for various domains, making them a reasonable baseline for tasks outside image classification [3,4].
>     - *XY model representation*: The XY model can be visualized as a heatmap (Figure 3), which has a smooth structure similar to natural images. This makes ResNet18 a relevant baseline to test how well a general-purpose model can perform on this dataset.
>
>     We will clarify this choice in the revised manuscript.
>
> 9. **Clarification on Embedding (L240)**: The term “embedding” refers to the higher-dimensional space in which the manifold is embedded. We will clarify this terminology in the revised text.
>
> We hope these responses address the reviewer’s concerns and are happy to incorporate any additional feedback. Thank you again for your detailed review, which has helped us identify areas for improvement.
>
> **References**
>
> [1] Taco Cohen, Mario Geiger, Maurice Weiler, “A General Theory of Equivariant CNNs on Homogeneous Spaces”
>
> [2] John Lee, “Introduction to Smooth Manifolds”
>
> [3] Yuki Naga and Akio Tomiya, “Gauge Covariant Neural Network for Quarks and Gluons”
>
> [4] Luo et al., “Gauge-invariant and Anyonic-Symmetric Autoregressive Neural Network for Quantum Lattice Models”

---

### Official Review · Reviewer_a1CZ · 2024-11-03

**Soundness:** 4
**Presentation:** 3
**Contribution:** 3
**Rating:** 6
**Confidence:** 2

**Summary:**

This paper introduces a novel approach to incorporating gauge invariance in equivariant neural networks for physical systems. The authors develop a general architecture that explicitly models gauge fields within graph neural networks, allowing for the handling of arbitrary fiber bundles rather than being restricted to specific cases like tangent bundles. The method is evaluated on the XY model across different geometries (flat, torus, sphere), demonstrating superior performance in energy estimation and vortex classification tasks compared to baseline models.

**Strengths:**

1. The idea is novel and the authors provide a clear explanation of connections to existing works.
2. The paper provides a comprehensive theoretical foundation, carefully connecting differential geometry concepts to practical neural network implementation.
3. The model achieves loss ~10^-4 with only ~10^3 parameters, while ResNet requires ~10^7 parameters for loss ~10^-2 (Fig 2b). This represents a 100x improvement in accuracy with 10000x fewer parameters.

**Weaknesses:**

1. In its current form, the paper can be hard for non-physicists to decipher. It would be preferable to focus more on the intuitions in the main text and move some technical discussion into the appendix.
2. Although impressive, the model is only evaluated on the classical XY model. It would be interesting to explore other classical and quantum models.

**Questions:**

1. How does the choice of graph discretization affect the gauge field representation, particularly for curved manifolds like spheres?
2. How can the model be applied to quantum systems?

---

> ### Author Response · Authors · 2024-12-02
>
> We thank the reviewer for their thoughtful feedback and for appreciating our contributions. Below, we address the specific points raised.
>
> 1. **Clarity for Non-Physicist Audiences**: We agree that certain sections of the manuscript could be clearer for readers without a physics background. As also highlighted by Reviewer 4fJP, we are working on revising the manuscript to improve the exposition of the key ideas and reduce reliance on technical details in the main text. The revised version will shift some of the more technical discussions to the appendix while focusing on the intuitions and guiding principles of our approach in the main text.
>
> 2. **Application to Quantum Systems**: Thank you for this excellent suggestion. Extending our approach to quantum systems is an exciting direction we are actively exploring. However, identifying appropriate tasks and datasets has been challenging. For example:
>     - The Aharonov-Bohm Effect: One potential task involves modeling the phase shift of a quantum particle after passing through a double-slit setup near a solenoid. This requires encoding the wavefunction as a field and predicting measurable quantities like the phase shift. However, simulating wavefunctions with sufficient fidelity for learning tasks is computationally intensive.
>     - Electronic Systems with Topological States: Another direction involves studying electronic systems exhibiting topological states, which would require datasets generated through heavy simulations of these quantum systems.
>
>     We agree that demonstrating our model on such systems would further validate its utility and generality. While we have focused on the classical XY model in this work, we are designing experiments to apply our method to quantum systems and plan to present these results in an updated version.
>
> 3. **Impact of Graph Discretization**: Thank you for this question. One advantage of our approach is its independence from the specifics of graph discretization. By explicitly modeling the gauge transformation (i.e., how local frames transform between points) and treating the inputs as point clouds, our method adapts naturally to different graph topologies or discretization levels. This makes it particularly suited for curved manifolds like spheres, where the choice of discretization can vary widely. We will clarify this point in the revised manuscript.
>
> 4. **Additional Experiments on Classical/Quantum Models**: We appreciate the suggestion to explore additional classical and quantum models. While our current work focuses on the XY model, chosen for its simplicity and well-understood physical properties, we agree that evaluating our approach on a broader range of models would further strengthen its impact. For quantum systems, as mentioned above, the challenge lies in dataset preparation and computational overhead. However, we are actively pursuing these directions and hope to extend our results to both classical and quantum domains in follow-up work.
>
> We thank the reviewer again for their valuable feedback and hope these clarifications address their comments.

---

### Official Review · Reviewer_DvPx · 2024-11-03

**Soundness:** 4
**Presentation:** 3
**Contribution:** 1
**Rating:** 3
**Confidence:** 4

**Summary:**

The paper presents a generalized formulation of gauge equivariant networks that explicitly represents the gauge field, and works for arbitrary fiber bundles. The paper covers all the necessary background on gauge theory and fiber bundles, and then defines a GNN-based gauge equivariant architecture. The basic idea is to include an explicit gauge field as a map from edges to linear transformations of the fiber, and to use this to compute gauge in/equivariant features such as invariant inned products or bivector features. The paper then studies two application, energy estimation and vorticity estimation, and shows significant gains over non-gauge-equivariant methods.

**Strengths:**

+ The paper carefully explains background material in an accessible manner.
+ Well motivated
+ Somewhat novel formulation / architecture

**Weaknesses:**

- The paper lacks references to key papers, such as: "Gauge covariant neural network for quarks and gluons, Yuki Nagai & Akio Tomiya", "Equivariant Flow-Based Sampling for Lattice Gauge Theory, Kanwar et al.", "Learning lattice quantum field theories with equivariant continuous flows, Gerdes et al."

- The paper claims that the gauge field absent in earlier work. but this is not quite true. The works that use the tangent bundle generally make implicit use of the Levi-Civita connection. Other gauge equivariant networks exist that also use the connection before comparing features in different fibers (see e.g. the references above).

- Generally, even if some other works are specialized to particular kinds of fiber bundles, the paper does not present a truly novel idea for generalizing to arbitrary bundles. The paper simply takes the general idea of transporting features between fibers using a connection, and then says that one can do this in general (how to do this is standard stuff in the theory of fiber bundles).

- For most of the paper, it is not clear where the gauge field comes from. In eq. 20 it just says that A is a map from edges to fiber maps. Given that this is not an infinitesimal transformation, should it be computed by integrating or exponentiating? More generally, do I understand correctly that the gauge field should be derived from known theory? (I had hoped that the gauge field could be learned or computed as a feature)

- The paper does not do any ablations on the architecture, and compares to weak baselines without gauge equivariance. With the current evidence we can't tell if the particular features proposed in this work are really very effective or not.

**Questions:**

n/a

---

> ### Author Response · Authors · 2024-12-02
>
> We thank the reviewer for their thoughtful feedback and for pointing us to several references. While we are happy to include these works to provide a more comprehensive overview of related literature, we would like to clarify how our contributions differ from the cited references:
> 1. **Relation to the cited references**:
>     - “Gauge Covariant Neural Network for Quarks and Gluons” applies a smearing operation iteratively (referred to as a gauge-covariant map, Equations (8–14)), but it does not explicitly model the gauge field in a closed form.
>     - “Equivariant Flow-Based Sampling for Lattice Gauge Theory” proposes a gauge-equivariant coupling layer, constructed using gauge-equivariant kernels. However, the approach simplifies gauge theory by trivializing the gauge degrees of freedom rather than explicitly modeling the gauge field (as stated on p. 3, above “Application to U(1) gauge theory”).
>     - “Learning Lattice Quantum Field Theories with Equivariant Continuous Flows” explicitly states in the conclusion that their methods are not yet applied to gauge theories, suggesting this as future work.
>
>     We believe our contribution, which explicitly models the gauge field as part of the architecture, is distinct from these references. Our approach provides a unified framework for gauge-equivariant architectures applicable to arbitrary datasets, moving beyond task-specific designs or simplifications of gauge theories.
> 2. **Absence of the Gauge Field and Generality**: Our claims in the abstract (lines 14–19) and introduction (lines 82–92) highlight that existing work on gauge-equivariant architectures has been limited in *explicitly* modeling the gauge field. By explicitly parameterizing gauge transformations through matrices and working with graph data that are not restricted to square grids, our approach offers a general-purpose architecture that can adapt to diverse physical and machine learning problems.
> This explicit modeling of the connection allows for flexibility and applicability across a broad range of settings, which we believe is a significant contribution. While we agree that the concept of transforming features between fibers using a connection is standard in gauge theory, no existing work has implemented this in a general, data-agnostic manner suitable for arbitrary topologies and datasets.
>
> 3. **Gauge Field Origin**: We present a general framework where the gauge transformations are represented explicitly. Within our framework, gauge transformations can be "hard-coded" in problems where they are known, or they can be learned as part of the training process, providing an avenue for architectures to adapt to datasets and physical theories without the need for task-specific redesigns.
> We acknowledge the reviewer’s suggestion to demonstrate this more concretely and are currently running additional experiments to include in an updated version. These experiments aim to explore learning the gauge field from data.
>
> 4. **Ablations**: The core contribution of our work is a framework for building gauge-equivariant networks that generalize easily to new applications without requiring new mathematical tools or task-specific modifications. We appreciate the reviewer’s concerns about evaluating the effectiveness of our proposed features. To address this, we will include ablation studies in the updated version, comparing the architecture with and without the gauge corrections, to better quantify their impact.
>
> We hope this clarifies our contributions and how they relate to the cited works. Thank you again for your constructive feedback, which is helping us further strengthen our manuscript.

---

### Official Review · Reviewer_4fJP · 2024-11-04

**Soundness:** 2
**Presentation:** 1
**Contribution:** 2
**Rating:** 5
**Confidence:** 2

**Summary:**

The paper presents GaugeNet, an architecture designed to learn general gauge-invariant quantities by explicitly modeling the gauge field within a graph neural network framework. GaugeNet addresses the limitation of existing gauge-equivariant neural networks that are often constrained to specific cases, such as tangent bundles or quotient spaces, limiting broader applications to diverse gauge theories in physics. The authors validate GaugeNet on classical physical systems, such as the XY model on various curved geometries, demonstrating its effectiveness in capturing gauge-invariant properties.

**Strengths:**

* The approach is well-motivated, as developing a gauge-equivariant neural network adaptable to various fiber spaces is essential for capturing the full scope of gauge theories in physics. By explicitly modeling the gauge field, GaugeNet represents a meaningful step in this direction.
* GaugeNet shows excellent performance in handling gauge invariance in the XY model compared to conventional non-equivariant and equivariant neural networks, achieving this in a parameter-efficient manner.

**Weaknesses:**

* The methodology/theory section is challenging to follow, especially for a general machine learning audience. Several methodological aspects could benefit from additional clarification:
     * The connection between energy forms in discrete and continuous physical spaces is somewhat unclear. Specifically, why does the coupling matrix $J$ in Equations 1 and 2 disappear in Equation 4? Could the authors provide a step-by-step derivation of how Equation 4 is obtained from Equations 1 and 2?
    * The logic from lines 227 to 234 is hard to follow and could benefit from more elaboration.
    * The purpose of the “Curvature from Parallel Transport” section in establishing the gauge-equivariant framework is unclear. Could the authors include a brief explanation of how the "Curvature from Parallel Transport" section relates to the proposed method
    * Are there any variations in the physical forms of the field value  $\mathbf{s}_i$  across different topologies? If so, does GaugeNet need architectural adjustments to accommodate these cases? Additionally, why are all field values represented as vector features instead of higher-order features?
    * What distinguishes $\mathbf{s}_i$  from  $\mathbf{x}_i$ in Equation 23? If $\mathbf{x}_i$ represents vector features of the field, does the term  $\mathbf{x}_i - \mathbf{x}_j$ maintain equivariance to local gauge transformations?
* The comparison of GaugeNet is limited to two global equivariant neural networks, EMLP and EGNN, rather than models that address local equivariant transformations, as mentioned in the literature review (e.g., [1], [2]). This hinders readers' justification for the advantage of GaugeNet over previous methods in this area.

[1] Pim de Haan, Maurice Weiler, Taco Cohen, and Max Welling. Gauge equivariant mesh cnns: Anisotropic convolutions on geometric graphs.
[2] Di Luo, Zhuo Chen, Kaiwen Hu, Zhizhen Zhao, Vera Mikyoung, and Bryan Clark. Gauge-invariant and anyonic-symmetric autoregressive neural network for quantum lattice models

**Questions:**

* Could the authors provide insight into why EGNN struggled to converge to a reasonable solution in the vortex numbers classification task?
* Possible typos:
    * Line 431: Does the first “equivariant” refer to “non-equivariant”?
    * Line 437: The sentence appears incomplete.

**Details Of Ethics Concerns:**

No ethics concerns.

---

> ### Author Response · Authors · 2024-12-02
>
> Thank you for the thoughtful feedback and for highlighting areas where our presentation can be improved. We appreciate the constructive suggestions and have addressed the main comments below. We are also revising the sections identified as unclear.
> 1. **Clarifying Equation (4)**: We appreciate your observation regarding the coupling matrix $J_{ij}.$ To clarify:
>     - In the discrete setting, $J$ represents the adjacency matrix of a weighted graph where interactions occur. This graph may or may not correspond to a discretization of a continuous space.
>     - When discretizing a continuous space model, $J$ contributes to the metric of the base space. For example, if the continuous model’s energy functional is given by:
> $\int dx^2 \lVert D S(x)\rVert_\eta^2 = \int dx^2 \eta_{\mu\nu} (D_\mu S(x))^\dagger D_\nu S(x),$
> then in the discrete case, $D_\mu S$ is approximated as:
> $D_\mu S \approx \frac{(1 + A_{ij})S_i - S_j}{\delta x_{ij}},$
> and the corresponding term transforms to:
> $\eta_{\mu\nu} \frac{\|(1 + A_{ij})S_i - S_j\|^2}{\delta x_{ij}^2}.$
> Assuming $\eta_{\mu\nu} = f_\mu(x) \delta_{\mu\nu}$, the metric simplifies to $J_{ij}$ with $f_\mu(x) = \sqrt{J_{ij}} / \delta x_{ij}$. We will incorporate this derivation into the paper to better connect Equations (1), (2), and (4).
> 2. **Logic in Lines 227–234**: We are including the full derivation of Equation (8) in the revised paper. Briefly:
>     - Gauge-invariant systems allow local features to be expressed in arbitrary bases without affecting measurable quantities like energy. However, when derivatives are involved, partial derivatives are not invariant under local (gauge) transformations.
>     - Covariant derivatives $D = d+ A$ include the gauge field $A$, which transforms as $A \mapsto A’ = gAg^{-1} -(dg)g^{-1}$ (Equation 8). Gauge-invariant quantities remain unaffected by such transformations, ensuring consistency across different bases.
>     - From Equation (7), $D = d+ A$. For gauge equivariance, $D’(gf) = gDf$ implies:
> $d(gf) + A'(gf) = g(df + Af) \implies dg \cdot f + g\cdot df + A' gf = g\cdot df + g Af\implies dg \cdot f + A'gf = g Af$
>
> This implies $dg + A'g = gA\implies A' = gAg^{-1} - (dg)g^{-1}$.
>
> 3. **Curvature from Parallel Transport**: Thank you for pointing out the lack of direct use of the curvature $F$ in our architecture. We included this section to demonstrate how gauge-invariant quantities can be derived from $A$. For instance, $F$ transforms as $F \mapsto gFg^{-1}$, simplifying construction of gauge-invariant terms compared to $A$. For example, the electric and magnetic fields in electromagnetism are components of $F$. We will move the section to the appendix to streamline exposition.
>
> 4. **Variations in Field Representation Across Topologies**: We agree this is an important point. Variations do occur across topologies, corrected by gauge transformations. For example:
>     - In the XY model, features are always two-dimensional vectors. However, for spheres and tori, these are embedded in three-dimensional space.
>     - Our architecture explicitly represents features as vectors, but this is not a limitation. Higher-order features can be flattened into vectors using irreducible representations. For instance:
> $\textrm{vec}(\rho_2(g)M\rho_1(g^{-1})) = \rho_2(g) \otimes \rho_1(g^{-1})^T \textrm{vec}(M),$
> as [1] and EMLP. We will include this explanation and relevant citations.
> 5. **Distinguishing $s_i$ and $x_i$ in Equation (23)**: Thank you for noting this ambiguity. In Equation (4), $x_i$ are base space coordinates, and $S(x_i)$ are feature values. As coordinates, $x_i$ do not affect gauge equivariance properties of the features. It is possible to have group actions on the coordinate spaces, but they are separate from those on fibers. We will clarify this distinction.
> 6. **Comparisons with [1] and [2]**:
>     - [1] assumes gauges are aligned and operates on rectangular grids with clear neighbor relations. This restricts its applicability to general graphs, unlike our method.
>     - [2] focuses on discrete quantum lattice systems. Extending it to arbitrary topologies would require generalizing its “composite particles,” which seems non-trivial and outside its current scope. Our framework, by contrast, handles continuous gauge theories and diverse topologies, offering broader applicability. We will elaborate on these distinctions in the revised paper.
> 7. **EGNN Performance in Vortex Classification**: EGNN struggled to converge, with training accuracy fluctuating around random guessing. We speculate this is related to how the node features are combined into the vortex estimation.
> 8. **Typos**:
>     - Line 431: “Equivariant” is correct, referring to EMLP.
>     - Line 437: Thank you for catching this; the sentence was truncated during editing.
>
> We hope these clarifications address the concerns raised and are happy to elaborate further. Thank you again for your valuable feedback.
>
> [1] Geiger, Smidt, "e3nn: Euclidean Neural Networks".

---

### Meta-Review · Area_Chair_GVUX · 2024-12-17

**Metareview:**

This paper presents methods for incorporating gauge-invariance in equivariant networks. Unfortunately, the reviewers raised a number of issues about it, including that the methods are very hard to follow and not well-written, and the comparison to several baselines is lacking, some of the experiments are poorly motivated (e.g., why is a general-purpose ResNet chosen as a baseline when it is not designed for graphs? is there no more appropriate method?), and that the explanation of contributions relative to prior works is lacking and unclear. It will be better for the community to address these issues before publication.

**Additional Comments On Reviewer Discussion:**

While reviewers appreciated the motivation, theoretical grounding, and potential impact of the work, they raised concerns about the clarity of exposition, limited comparisons to other gauge-equivariant methods, and the lack of ablation studies and validation on diverse datasets. The use of weak general-purpose baselines like ResNet18 also detracted from the paper's strength.

---

### Decision · Program_Chairs · 2025-01-22

Reject